# Beyond Visual Confusion: Understanding How Inconsistencies in ENS Normalization Facilitate Homoglyph Attacks

## ABSTRACT

In recent years, the Ethereum Name Service (ENS) has garnered significant attention within the community for enabling the use of Unicode in domain names, thereby facilitating the inclusion of a wide array of character sets such as Greek, Cyrillic, Arabic, and Chinese. While this feature enhances the versatility and global accessibility of domain names, it concurrently introduces a substantial security vulnerability due to the presence of homoglyphs—characters that are visually similar to others across Unicode and ASCII sets. These similarities can be exploited in homoglyph attacks, posing a distinct threat to domain name integrity. Despite community efforts to counteract this issue through a normalization process prior to domain resolution, our analysis uncovers significant discrepancies in how the normalization processes are applied across various applications. This inconsistency could result in the same domain name being resolved to different addresses in different applications, underscoring a critical vulnerability. We also discovered the new attack scenario in ENS which may cause legitimate domains resolved into malicious addresses even when they are verified by authorities. To systematically evaluate this inconsistency, we designed a tool for detecting application-level discrepancies in domain normalization process without requiring access to the application's source code. Our evaluation on hundreds of real-world Web3 applications identifies widespread deviations from established homoglyph mitigation practices, with more than 60% digital wallets and 80% dApps (decentralized applications) not able to produce consistent ENS resolving results, potentially impacting millions of users. This analysis underscores the urgent need for a standardized implementation of normalization processes to safeguard the integrity and security of ENS domains.

**ACM Reference Format:**
Anonymous Author(s). 2024. Beyond Visual Confusion: Understanding How Inconsistencies in ENS Normalization Facilitate Homoglyph Attacks. In . ACM, New York, NY, USA, 11 pages. https://doi.org/10.1145/nnnnnnn.nnnnnnn

## 1 INTRODUCTION

The Ethereum Name Service (ENS) operates as a decentralized, accessible, and adaptable naming system that leverages the Ethereum blockchain. Its primary function is to associate human-readable names like 'alice.eth' with machine-readable designations, which

encompass Ethereum addresses, addresses for other cryptocurrencies, content hashes, and additional metadata [12]. Over the past few years, ENS has garnered significant attention within the community. ENS employs Unicode encoding, with the flexibility that enables the encoding of an extensive character set, encompassing characters from diverse scripts such as Greek, Cyrillic, Arabic, and Chinese. The utilization of Unicode characters in domain names also facilitates the incorporation of regional alphabets into domain naming conventions.

However, a major security risk is introduced along with Unicode characters. The Unicode system contains characters that are visually similar to other Unicode or ASCII characters, called homoglyphs. A homoglyph attack is one technique for carrying out this scheme. For example, using most sans-serif fonts, the Latin letter l (lower case 'el') is visually confusable with the Latin letter I (upper case 'eye'). Rendered with such a font, the following domains are confusable:



**paypal̲.com** vs. **paypaI̲.com**



An attacker who owns the homoglyph domain name, paypaI̲.com, therefore may be able to lure victims to send transactions to their wallets/contracts, for example by sending a scam that appears to contain a link to the original paypal.com.

Homoglyph attacks have existed for years. In Domain Name System (DNS), the adoption of International Domain Names (IDNs) support also introduced the same problem [17]. Many international letters have similar glyphs. Due to the potential abuse of homoglyph characters, browser vendors have been exploring techniques to mitigate the homoglyph attack [10][26]. In ENS, the community has been aware of the problem and since December 2021, the community has decided to enforce the normalization process before domain resolution to mitigate the threat [28]. In the past two years, the normalization standard has evolved to cover the majority of homoglyph characters in Unicode. However, not all applications are following the standard closely. Many applications are still using the first version of the normalization or even not containing a normalization process at all.

While there are numerous parallels, homoglyph attacks within the ENS present a more severe risk than those encountered in the DNS. In DNS, homoglyph attacks often involve deceptive content designed to trick victims into divulging sensitive information to the attacker's servers. Throughout this process, victims may encounter multiple cues that could alert them to the fraudulent nature of the interaction, as illustrated in Figure 1. Indicators such as outdated website information, inaccuracies in product listings, or failures in input validation may serve as warnings. Moreover, even if a DNS-based homoglyph attack succeeds, the repercussions of leaked sensitive data can often be mitigated; for instance, compromised credit cards can be frozen, or passwords for breached accounts can be changed. Conversely, in ENS, the act of directing a transaction to a homoglyph domain constitutes the entirety of the attack, as depicted in Figure 1. Unlike in DNS scenarios, where victims might

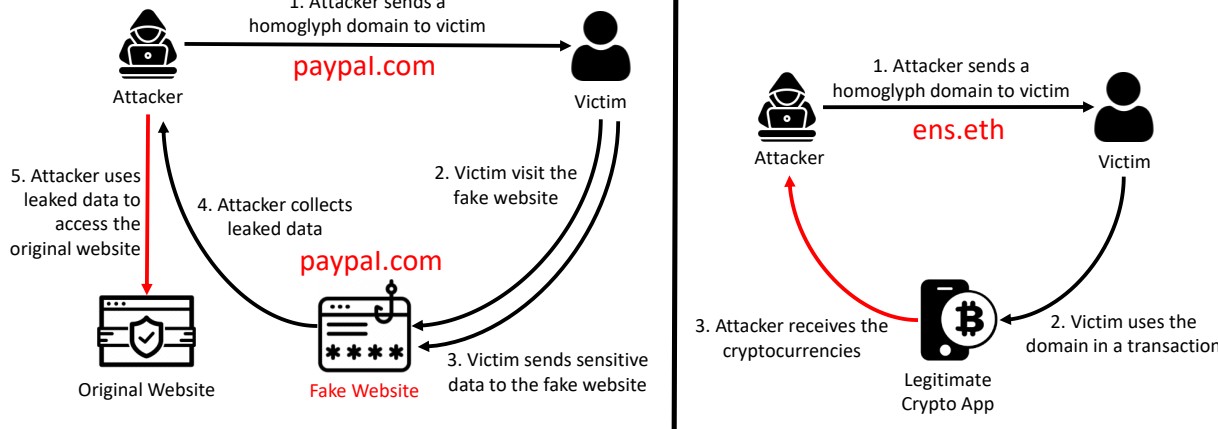

Figure 1: Homoglyph Attacks in ENS and DNS. The red lines are the steps where attackers achieve the attack goals. Victims are able to stop the attack before that. In DNS, the victim needs to perform two operations to be attacked and the attack effects are recoverable in some cases. However, in ENS, the victim only needs to type in the homoglyph domain in the crypto app they trust and the attack effects are almost always unrecoverable.

scrutinize the content of a fraudulent website, individuals targeted by ENS homoglyph attacks are less inclined to verify the legitimacy of the domain's associated address. Furthermore, the consequences of falling victim to a homoglyph attack in ENS are invariably financial losses, which, due to the immutable nature of transactions on the Ethereum blockchain, are irrevocable.

In this paper, we seek to investigate the extent of possible homoglyph risks in ENS and how well mitigation measures are enforced in the wild. The main contributions of this paper are as follows.

- We systematically analyzed the design of ENS normalization, identified the bad practices that lead to the inconsistencies in libraries and dApps, and discovered new attack scenarios where legitimate domains may also be affected.
- We propose a novel approach to automatically detect inconsistencies in the normalization process of an application compared to the latest normalization standards.
- We measured the inconsistencies in the implementation of ENS normalization processes in popular Web3 libraries, wallets, and dApps. As a result, we identified 214 dApps, 41 crypto wallets, and 11 Web3 libraries that are inconsistent with the official homoglyph mitigation and could impact millions of users.

The remainder of this paper is organized as follows. In Section 2, the background of homoglyph attacks and ENS is given. In Section 3, the measurement of homoglyph domains in ENS is presented. Inconsistencies in the normalization processes are provided in Section 4. Section 5 discusses related work in the literature. A discussion around drawbacks that still need to be addressed is given in Section 6. Finally, Section 7 concludes the paper.

## 2 BACKGROUND AND MOTIVATION

In this section, we first introduce the background of ENS. Then we motivate our work with the new threats in ENS.

### 2.1 Ethereum Name Service

ENS shares common objectives with DNS, although it diverges significantly in architecture due to the unique capabilities and constraints offered by the Ethereum blockchain. Much like DNS, ENS functions using a structure of hierarchical names separated by dots, referred to as domains, where the owner of a domain wields complete authority over its subdomains[12]. Smart contracts known as registrars hold ownership of top-level domains, such as '.eth,' and establish regulations that govern the distribution of their respective subdomains. These registrar contracts lay out guidelines for anyone to acquire domain ownership by adhering to the specified rules. ENS also offers the flexibility to import DNS names that users already own, making them available for use within the ENS framework.

Due to ENS's hierarchical structure, individuals possessing a domain at any level are empowered to configure subdomains according to their preferences, whether for personal or external use. For example, if Alice is the owner of 'www.eth' she can establish '2024.www.eth' and customize it to her specific requirements.

The workflow of ENS is shown in Figure 2. The domain owner first registers a domain through an ENS controller (e.g., ENS official website and MyEtherWallet). The controllers will process the domain registration request and craft a domain registration transaction. The transaction will be sent to ENS registry contract on the Ethereum blockchain. Once the transaction is processed, the domain is registered. Then, the domain owner advertises the domain to his/her users. When the user tries to access the domain, the wallet or client that the user uses will craft an ENS domain resolving transaction and send it to the domain resolver contract on the blockchain. However, different from the registration transaction, the resolving transaction will not be submitted to the blockchain since the record is available on any node.

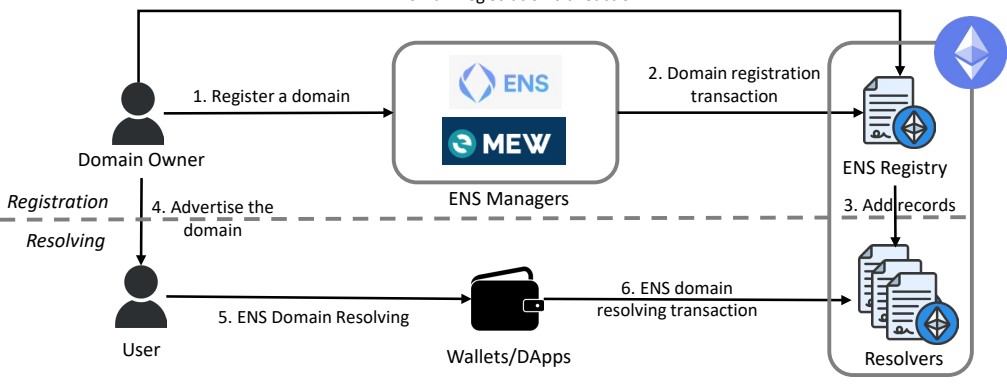

**Figure 2: ENS Workflow**

## 2.2 Motivation

In the ENS, domains are registered not by their apparent textual representation but as hashed values resulting from a normalization process. For instance, a domain like vitalik.eth is actually recorded on the blockchain in its hashed form, i.e., [0xee6c4522aab0003e8d14cd40 a6af439055fd2577951148c14b6cea9a53475835]. This characteristic significantly impacts the defense mechanisms against homoglyph attacks, relegating them primarily to application-level implementations. Unlike in the DNS system for IDNs, where validation is standardized and occurs through DNS servers because the domain names are encoded rather than hashed, ENS operates differently. The resolution process in ENS maps a hashed domain to a specific wallet or contract address without an intermediate step to validate the domain post-hashing. Consequently, the security against such attacks hinges entirely on how applications manage the normalization and hashing processes. This unique challenge within ENS security protocols has prompted our investigation into these application-level implementations.

On the other hand, considering the scenarios that ENS is used, homoglyph attacks in ENS have the potential to cause more severe security impacts to users than in DNS. In DNS, a homoglyph attack may lure the victim to a phishing website and trick them into sending sensitive data to the attackers. For example, a phishing website could be masked as an e-commerce website. During the homoglyph attack, the victim may notice that a) the domain is weird, b) the prices are not reasonable, or c) the website is not checking the validity of the card information. In other words, in scenarios involving sensitive data compromise, the victim typically has several opportunities to recognize the threat and halt the process before their data is transmitted to the cybercriminal. Even if the card details are dispatched, the victim retains the option to lock the card, significantly reducing potential damages. Contrastingly, within the ENS, the use of a homoglyph domain in any transaction results in immediate financial detriment to the victim, who has a singular opportunity to detect and avert the assault. More critically, financial losses incurred in such instances are often irreversible. Therefore, this paper evaluates the magnitude of this issue and examines the effectiveness of mitigation strategies, specifically the enforcement of the normalization process, in real-world settings.

## 3 OUR DISCOVERIES

ENS names have been integrated into hundreds of decentralized applications (dApps) and wallets that are the foundation of Web3. Hence, homoglyph domains in ENS presents a significant threat to the security of the Ethereum ecosystem. Since ENS has a wider support of Unicode characters than DNS, homoglyph domains in ENS can be more diverse and sophisticated. In this section, we measure the prevalence of homoglyph domains in ENS, identify the popular homoglyph characters, and characterize the homoglyph domains. We also present our findings on the new attack vectors and scenarios that homoglyph domains in ENS can introduce.

## 3.1 Understanding Normalization in ENS

In this section, we present new attack vectors and scenarios that homoglyph domains in ENS can introduce. We first present our preiminary study on the inconsistencies in the normalization process of ENS, which can be exploited by attackers to bypass the security checks. We then discuss the security implications of the inconsistencies and demonstrate them in real-world to show the impact.

*3.1.1 Normalization Process.* The ENS name normalization process standardizes the transformation of ENS names into a canonical form before they are hashed for registration, which is defined in ENSIP-15. ENSIP-15 extends ENSIP-1 by incorporating the latest Unicode standards and addressing modern challenges, such as emoji sequences and homoglyph attacks. Normalization ensures that a name is transformed into a consistent representation, even if input varies. The process consists of the following steps:

(1) **Tokenization:** The input is split into labels, which are further divided into sequences of Unicode code points. Each label is tokenized into Text and Emoji tokens.
(2) **Unicode Normalization:** Text tokens are normalized using the NFC (Normalization Form C) standard, which combines diacritical marks with base characters. Emoji tokens are simplified by stripping optional presentation characters (such as FE0F).
(3) **Validation:** The tokens are validated to ensure compliance with character sets allowed in ENS labels. This includes checking for disallowed characters, ensuring consistency

between input and output, and verifying that emojis follow standard sequences.

(4) **Concatenation:** The normalized tokens are concatenated into a string of Unicode code points, resulting in the normalized label.

*3.1.2 Inconsistency Types.* In this subsection, we explore three primary sources of inconsistencies in the implementation of normalization processes we found in our preliminary study as detailed in Appendix A. First, we discussed how discrepancies can emerge due to the use of different versions of the Unicode library, emphasizing the need for standardized, version-independent approaches to ensure consistent character handling. Second, we examine the tension between official library implementations and the Unicode Technical Standard #46 (UTS46), highlighting that variations can occur when applications opt for customized normalization routines based on the UTS46 standard. Lastly, we dive into the impact of customized normalization functions within applications, which can lead to deviations from standardized normalization outcomes. The key takeaway from this subsection is the importance of clear communication, documentation, and standardization in the Unicode normalization ecosystem to minimize these inconsistencies and facilitate smooth interoperability between diverse applications.

**Type 1. Different Versions of Predefined Rules.**

One of the prominent sources of inconsistency in the implementation of normalization processes stems from the use of different versions of the predefined rules. As the normalization standards evolve and the library is updated to accommodate these changes, discrepancies can emerge between applications that utilize distinct versions of the library. These disparities may manifest in variations in normalization results, which can lead to interoperability issues. For instance, a wallet relying on an older version of the predefined rule list may consider certain characters valid, while another application utilizing a more recent version might return error with the same character. As a result, these disparities in library versions can hinder the seamless exchange of data and content between different applications.

**Type 2. ENSIP-15 vs. UTS46 Standard:**

Another source of inconsistency pertains to the confluence of ENSIP-15 and UTS46. The ENS community announced that UTS46 is used as the standard to encode the domain names. However, UTS46 itself does not contain any normalization process. While UTS46 is a respected standard for the Unicode community, variations can arise when applications only process inputs based on this standard rather than adhering strictly to the official normalization process. Consequently, the application with only UTS 46 support is much more tolerant to homoglyph characters, which makes homoglyph attacks easier to the users.

**Type 3. ENSIP-15 vs. Customized Functions:**

Inconsistencies can also arise from the customization of normalization functions. While ENSIP-15 offer a standardized approach to ENS domain normalization, some applications may choose to implement customized normalization functions tailored to their specific needs. One of the major reasons we observed is that the official libraries only supported a limited number of programming languages, while the applications are written with wider choices. However, these customization can result in inconsistencies in the normalization results. It becomes crucial for developers to thoroughly document and validate these customization to ensure that the same rules are applied as defined in ENSIP-15.

## 3.2 New Attack Scenarios

Different from homoglyphs in other fields like DNS, where a banned domain will get invalid in a few minutes all over the world, in ENS, due to its nature that the normalization is proceeded off the chain, any domain that has been registered can still be resolved even when it is not allowed in the normalization standards any more. With such feature, the inconsistency issue can cause severe security implications to users. In this section, we further analyze the security implications of the inconsistencies and demonstrate them in real-world to show the impact depending on how the character is handled by the standard normalization process.

**Disallowed Homoglyph Characters.** If a character is not allowed (i.e., a standard normalization process will raise an alarm and stop the resolving), the domain is considered not valid in the community. Mostly, a character is not allowed because the community feels the character is used by attackers more than regular users [14]. However, in the current ENS ecosystem, such domains can still be resolved by Crypto applications that are using inconsistent normalization libraries. Hence, the users of those applications are under the risk of phishing/scam attacks though an effective (standard normalization) mitigation is provided by the community.

**Ignored Characters.** Specifically, some formatting characters and special characters used for emoji sequences (e.g., U+FE0F) will be ignored during the normalization process, while domains containing them are still considered valid in the community. In this case, a domain containing such characters can still be accepted and resolved by all applications. However, since these characters are zero-width, they can be removed or placed in different positions of a emoji sequence when the emoji is rendered exactly the same in some fonts. Hence, if an attacker registered a domain with ignored characters on the chain, a vulnerable application will resolve both the attack domain and original domain into the attacker's address. For example, in Figure 3, the original domain only contains three wallet icons, where U+FE0F is just for emoji sequences. With a standard normalization implemented by authorities like Etherscan[15], U+FE0F will be ignored and the domain in Figure 3 will be resolved into the original address. However, a legitimate Crypto application with inconsistencies in normalization process may not ignore U+FE0F and try to resolve the domain with it. Finally, the application will use the one registered by the attacker for later transactions, which is not what the user intended at the beginning.

It it worth noting that with the new attacks, even if the user has concerns on the domains and verifies them on authorities like Etherscan[15], they may still be attacked because authorities are strictly following the latest official normalization standard while the dApps may not. During our evaluation, we have identified that even popular wallets like MathWallet, with more than one million users, are still vulnerable to this type of attacks. The beautifier proposed recently [29] worsened the issue by always adding formatting characters back to the domain for a better display, which has been deployed by many popular websites and applications.

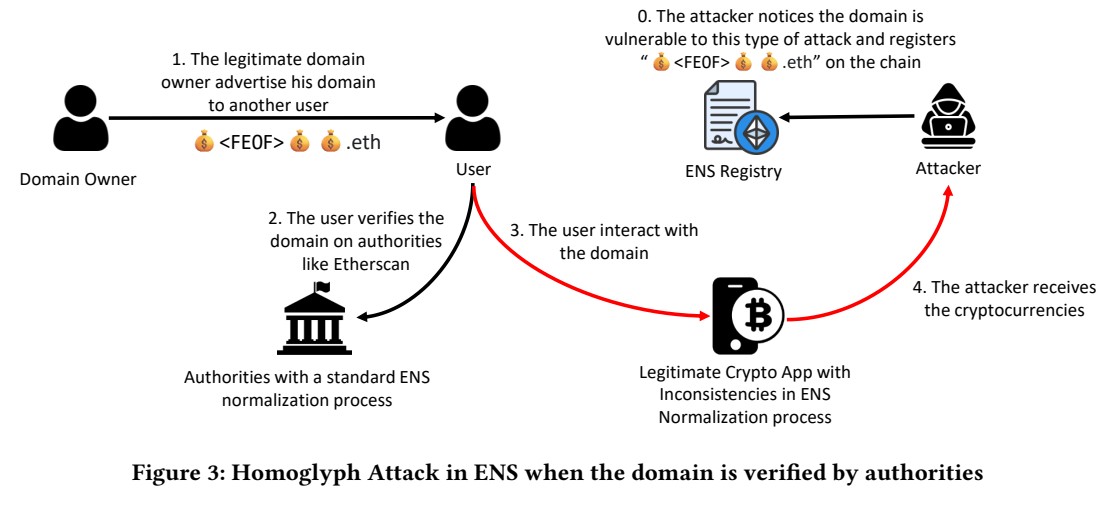

**Figure 3: Homoglyph Attack in ENS when the domain is verified by authorities**

As shown in Figure 3, with the new attacks, even if the user has concerns on the domains and verifies them on authorities like Etherscan, they may still be attacked because authorities are strictly following the latest official normalization while the dApps may not.

## 4 MEASURING INCONSISTENCIES IN NORMALIZATION PROCESSES IN THE WILD

Since the ENS community has been aware of homoglyph attacks, and ENSIP-15 has been proposed as a mitigation against the attack. In this section, we seek to answer the question:

- Is the normalization defined in ENSIP-15 correctly enforced in different crypto applications?

### 4.1 Data Collection

The evaluated applications are collected as follows. We collected all listed apps on ENS official website under ENS ecosystem. In Web3, the community shows great interests of making mobile as the mainstream platform [2]. The report from MarketGrowth mentioned that significant growth of mobile crypto wallet market is expected in the next decade [1]. The convenience provided by the mobile platform also brings risks. We noticed that almost all browser extension wallets and web version dApps will provide the resolved address of the input ENS domain, while on mobile, this is on the contrast. Hence, in this paper, we focus on the mobile platform as with less information, users are likely to be attacked if the app does not enforce the normalization correctly. Finally, we collected 13 libraries, and 264 dApps (including 67 wallets) for our evaluation.

### 4.2 Automatic Normalization Inconsistency Detection Approach

To automatically identify inconsistencies in ENS normalization within applications, we developed a tool. This prototype system is designed to analyze and understand the handling of ENS-related operations within mobile applications, as illustrated in Figure 4. The design of each component is detailed below.

**Trace Collector**. The process begins with instrumentation, recording all function calls during the application's runtime in execution traces. As users engage with ENS-related functions, the tool captures the triggered function calls, focusing on those relevant to ENS normalization. The function names are extracted from the smali code of the APK file of the target Android application. Using this function list, we hook the function calls with the help of Frida [16], capturing both the function calls and their return values. In this paper, we hooked all the function calls. Sometimes this will cause significant performance overhead during the user interactions in the next step. However, since the hooking script can be added to the runtime after everything is setup for ENS-related operations, during our testing, only a few seconds latency are introduced on average, which is reasonable for general testing. Specifically, Dapps that use decoupled design pattern are exceptions, like Coinomi, for which we waited several minutes before we caught the ENS-relevant function calls. As such Dapps is a minority in our dataset (7 out of 264), we plan to address this issue as our future work.

**Trace Analyzer**. Automatically identifying ENS-relevant function calls within applications presents a significant challenge due to the vast number of function calls that must be scrutinized. Our method simplifies this complexity by specifying input values, such as "test.eth," and then rigorously examining each function call and its return values. We specifically look for instances where a function call includes an argument that matches our predefined input value and yields a return value that is the resolved address of the domain. When these criteria are met, we classify the function as an ENS-resolving function. Within the ENS-resolving function, the tool searches for function calls with "test.eth" as input and output, which are considered candidates for ENS normalization functions. By dynamically calling these candidates with a domain containing different character sets, the tool can further determine which candidate is indeed an ENS normalization API. For each identified API, the tool generates a Frida script to automatically test the API with a given set of domains to check for inconsistencies in the normalization process.

A further challenge in identifying inconsistencies arises when some applications rely on server-side APIs, which impose strict rate limits. To address this, we developed a library probing technique that identifies the normalization library used by the server-side API with as few as 4 and up to 13 requests for the evaluated libraries.

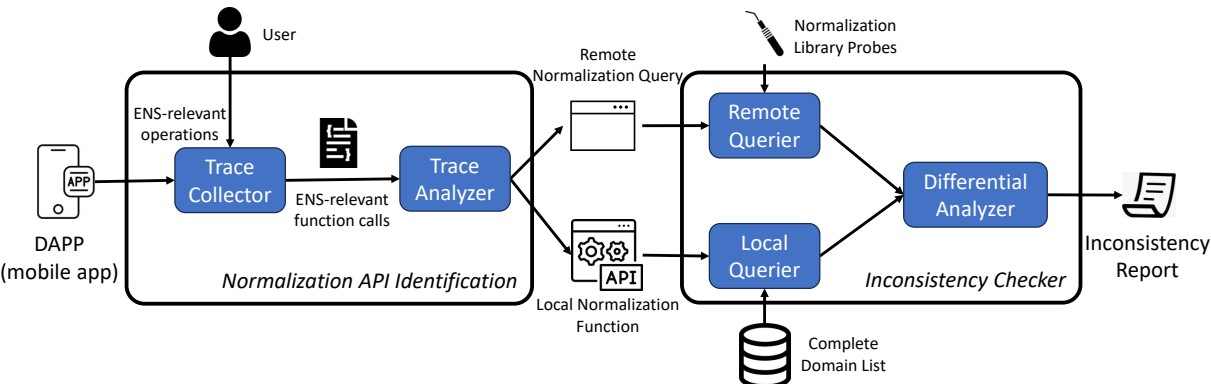

**Figure 4: Inconsistency Detection Workflow**

Our library probing technique rigorously assesses normalization outcomes for each domain in our dataset across multiple libraries, comparing results to distinguish between libraries. Differentiation is based on analyzing error messages and examining characters that are altered or omitted from the inputs. Importantly, our method efficiently identifies the most possible server-side library implementations without triggering rate limits, allowing identification with as few as one request. This precision and efficiency mitigate potential rate-limiting concerns while providing a nuanced understanding of server-side processing and the distinctive characteristics of each library's approach to domain normalization.

To determine whether a specific API is a remote API, we employ a latency-based approach. Prior to testing, the network is proxied, and all responses from the server are delayed by 2 seconds. If the API is a remote API, this delay will be reflected in the execution time of the API, as local APIs are not affected by network latency.

**Local/Remote Querier**. With the identified API, we can query it with different domains to check for inconsistencies in the normalization process. The tool automatically generates a Frida script to test the API with a set of domains. Results are collected and analyzed to identify inconsistencies in the normalization process. For local APIs, the tool directly calls the API with the given domain, using the complete domain list from the ENS official database [4]. For remote APIs, we use the previously identified library probes to query the server-side API. With the minimum query set (4-13 domains for our evaluated libraries), we can efficiently identify the normalization library used by the server-side API. Even if the server does not exactly use the identified library, the results remain valid, as normalization outcomes are consistent across our tests, which could still be exploited by attackers.

**Differential Analyzer**. Finally, the tool analyzes the results from the local/remote queries to identify inconsistencies in the normalization process. It generates a report outlining these inconsistencies and providing insights for improvement. This report serves as the basis for identifying potential issues, as demonstrated in Section 3.

The culmination of this testing is a detailed report that outlines the application's handling of homoglyph characters and normalization processes, identifies deviations from standards, and offers insights for improvement. This tool significantly benefits developers and quality assurance teams by providing a thorough analysis that highlights areas for enhancement, thereby ensuring character handling consistency, enhancing interoperability, and improving the overall user experience.

By streamlining the process of detecting normalization inconsistencies and providing actionable insights, this tool represents a valuable asset for ensuring the robustness and reliability of ENS-supported mobile applications. It offers a proactive approach to identifying and rectifying potential issues, saving time and resources while contributing to the development of more dependable and user-friendly applications.

**Table 1: Inconsistencies in ENS Normalization Processes**

| Type of Application | Total Count | Found Inconsistent |
|---|---|---|
| Libraries | 13 | 11 |
| Wallets | 67 | 41 |
| dApps | 264 | 214 |

## 4.3 Inconsistencies in Crypto Applications

We sought to investigate the ENS, Ethereum Name Service, normalization processes across a diverse array of applications, including dApps, wallets, registrars, and libraries. To answer the research question, we need to uncover and document inconsistencies rendered in the ENS normalization process of these applications.

Our comprehensive assessment encompassed twenty cryptocurrency wallets, revealing notable findings as detailed in Table 1. Wallets play a pivotal role in cryptocurrency transactions by securely managing public and private keys and facilitating interactions across various blockchains. Our thorough examination uncovered that 70% of the wallets evaluated, equating to fourteen out of twenty, exhibited inconsistencies in their ENS normalization processes. Such discrepancies could potentially lead to significant implications, posing challenges in user interaction and transaction processes, especially considering their extensive user base as illustrated in Table 2. Since there is no exact number of app users available, we take the following steps to estimate it. First, if the app

is available on Google Play Store, we use the number of downloads as the estimated number. Second, we search on Google and if the app owner has a claim on the number of app users, we take the number as the estimated number.

Table 2: Inconsistencies in normalization in selected evaluated Apps

| App Name | Inconsis. Type | # of app users | Verifiable Info |
|---|---|---|---|
| Torus | Type 3 | N/A | None |
| Burner | Type 3 | N/A | Address(partial) |
| OwnBit | Type 1 | >10k | None |
| Frontier | Type 1 | 50k | None |
| Coinomi | Type 1 | >1m | None |
| MyEtherWallet | Type 3 | ~1.3m | Address |
| Argent | Type 1 | >20k | None |
| Rainbow | Type 2 | >20k | Address |
| MyCrypto | Type 2 | N/A | Address(partial) |
| D'CENT Wallet | Type 2 | >20k | None |
| DexWallet | Type 3 | >10k | None |
| Math Wallet | Type 3 | ~1m | None |
| AlphaWallet | Type 2 | >50k | None |
| Ambire Wallet | Type 3 | >1k | None |
| Status | Type 2 | >1m | Address(partial) |

The study was also extended to two registrars, which are fundamental components in the ENS domain assignment protocol. The evaluation results were alarming, with a 100% inconsistency rate. Both registrars examined presented inconsistencies in their normalization processes, outlining a critical vulnerability that could potentially impact the efficient functioning of the Ethereum Name Service.

Reviewing the libraries was another major facet of our evaluation. Libraries are a vital resource for developers, offering pre-written codes, classes, procedures, scripts, configuration data, and more. They facilitate efficient coding and prevent reinvention of the wheel for common tasks, making them a lynchpin of efficient development work. In our study, 13 libraries were evaluated for possible inconsistencies in ENS normalization. Unfortunately, the results were disconcerting, with around 61% of libraries, 11 out of 13, presenting inconsistencies.

However, the most astounding results came from the evaluation of dApps. As the normalization process in dApps is crucial for ensuring efficient functionality, it was essential to check for any inconsistencies. The results were concerning; 214 out of 264 evaluated dApps were found inconsistent with the official normalization, an unignorable percentage that portrays a dire need for a review and revamping of the current ENS standardization process.

Taken together, these results underscore the prevalent inconsistencies that are spread over different aspects of the ENS normalization process. They highlight a substantial need for enhanced ENS normalization standards and practices across various Ethereum related applications. It is crucial not only to identify these existing inconsistencies but also to devise ways to rectify the anomalies detected in the process. It is worth noting that inconsistent normalization in the ecosystem is even more dangerous than no normalization at all. When inconsistent normalization presents, the threats will be more severe when users switch their app, exchange domains with their friends, posting on social media, and etc. As a result of these findings, our future work suggestions lay the foundations for rectifying these issues and propose effective optimization and standardization strategies for the ENS normalization process.

## 4.4 Case Study

In this section, we delve into two compelling case studies to illustrate how the inconsistency may affect the users.

**MathWallet.** Mathwallet is a popular crypto wallet application that supports more than 160 different blockchains, which is claimed to have more than 1 million users. However, in our evaluation, we found that its Android App resolves ENS domains by sending the query to the server side. With our tool, we queried the server with domains that are normalized differently in different libraries. The results show that the server side is likely to use the `eth-ens-namehash` as its normalization library. However, there is a huge gap between the normalization process in `eth-ens-namehash` and the normalization process in the official library. Even if a domain is malformed with homoglyph characters like zero-width joint character (U+200D), which is invisible, the domain can still be successfully resolved into the registered address on the chain, violating ENSIP-15. An attacker can abuse these characters to create a domain looks exactly the same as the target domain to lure the victims sending tokens to their addresses. Even worse, there is no way for the victims to validate it since the app will not show the resolved address to the user.

## 4.5 Responsible Disclosure

In our investigation, we identified 255 decentralized applications (dApps) and wallets with inconsistency issues. We attempted to reach out to the developers of these projects to responsibly disclose our findings. However, we were able to find valid contact information for only 207 of them. Among these 207 dApps and wallets, 192 lacked contact information specifically designated for security issues, complicating our efforts to communicate the security issues effectively. Despite our outreach, only 34 of the contacted entities responded, and of those, a mere 11 took corrective actions to address the issue. One of the responses from a developer provided insight into their reluctance to fix the inconsistency, stating that they believe it is the users' responsibility to verify domains before interacting with them. This response highlights a critical challenge in the ecosystem, emphasizing the need for better security communication channels and the importance of shared responsibility between service providers and users to enhance overall security.

## 5 RELATED WORK

In this section, we review related work on typosquatting attack and homoglyph detection.

## 5.1 Typosquatting Attack

Typosquatting stems from "domain squatting", a practice where individuals register domain names to sell them at a premium to their rightful owners later [33]. It extends this concept by exploiting human mistakes and the affordability of domain registrations to divert traffic from established websites [23], [31]. By registering domain names that closely resemble those of well-known sites, the

strategy banks on users making accidental typographical errors, thereby redirecting them to fraudulent sites instead of their intended destinations. The pioneering systematic investigation into this phenomenon was conducted by Edelman et al. [23], with subsequent research typically focusing on either measuring the impact on users [18] or identifying and analyzing Typosquatting strategies [32].

A critical aspect of addressing typosquatting involves understanding the criteria typosquatters use to select potential domain name typos. In this context, Mohaisen et al. [31] offered an in-depth review of prior research, which examines the various strategies employed by typosquatters in registering domain names. Some studies investigate specific forms of typosquatting, like those based on homophones [24] or binary bit flips [25], while others focus on creating a "squat space" through lexically similar URLs [31], with the latter being more relevant to our study.

The generation of squat spaces around popular websites typically relies on identifying domain names within a specific lexical range [31]. The Damerau-Levenshtein distance, which calculates the number of edits required to change one string into another, is frequently used in typosquatting to create these spaces [20].

Building upon existing studies on typosquatting, our study delves into the unique challenges and security considerations posed by homoglyph attacks in ENS. This connection between traditional typosquatting and our focus on homoglyph attacks in ENS underscores the evolving nature of cyber threats and highlights the importance of developing robust defense mechanisms tailored to the context of ENS.

## 5.2 Homoglyph Detection

A few approaches have been proposed to detect homoglyph domains in DNS. Yazdani et al. [36] combined DNS records and domain string characteristics to assess the homoglyph domains. Liu and Stamm [22] detect Unicode Obfuscated messages with the UC-SimList. Alvi et al. [9] focus on the obfuscation in plagiarism caused by Unicode characters. Their method uses the 'Unicode Confusables' list from Unicode community and the hamming distance. Krammer et al. [19] and Al Helou et al. [8] improved user interfaces for browsers to alert users for potential phishing attacks. Although alerting when Unicode is detected is useful in DNS and applied by many ENS applications, it is less useful in ENS as Unicode is much more common than in DNS.

Several studies, including those by Chiba et al. [11], Liu et al. [21], and Sawabe et al. [30], have explored the identification of homoglyphs that pose a threat to prominent brand domains by assessing the visual similarity of domain name images. While image-based methods eliminate the need for a homoglyph table, they have inherent limitations, primarily safeguarding a select set of domains, typically brand-related, and incurring significant computational costs when extended to the entire namespace. In contrast, Elsayed et al. [13] have developed a technique to identify potentially malicious domains within newly registered Unicode domains under the '.com' and '.net'. This method involves substituting Unicode characters with their corresponding ASCII homographs based on the 'Unicode Confusables' list. Additionally, they employ WHOIS data to differentiate between domains with malicious intent and protective domains. Quinkert et al. [27] have undertaken a similar approach, extracting homoglyphs that target the top 10,000 domains from the Majestic top 1 million domains [5], utilizing the 'Unicode Confusables' list as a reference. Xia et al.[35][34] looked into the security of ENS domain, focusing on squatting domains and domains detected malicious by VirusTotal.

## 6 DISCUSSION

The measurement results underscore the urgent need for a standardized approach to the normalization process across applications using ENS. The observed inconsistencies in normalization implementation highlight a fragmented ecosystem where the effectiveness of security measures varies significantly. This fragmentation not only undermines the security of ENS but also poses a risk to the overall integrity and user trust in decentralized naming systems.

**Limitations.** Recognizing the limitations of our study, particularly in our evaluation on the inconsistency of normalization processes, we observed that though the inconsistencies exist widely, there are minor differences between the inconsistency cases. For example, in the case study, we showed a domain with zero-width character, which is considered one of the most dangerous/confusing characters. However, some inconsistency cases are not that stealthy when there are specific input policies, (e.g., Upper case of i, I, to lower case of L, l when the input will automatically be converted into lower cases). In another word, besides the normalization process itself, the context in applications will also affect the effect of homoglyph attacks, including fonts, input policies, whether the resolved address will be shown or not, and etc. However, such context information can hardly be collected in an automatic way.

The potential for ENS domains to be rented through third-party smart contracts introduces another layer of complexity. This versatility, while a strength of the ENS system, complicates the task of monitoring and mitigating security risks. The exploration of third-party services and their impact on the security and usage of ENS domains is a promising avenue for future research, potentially uncovering new patterns of use or abuse that need to be addressed.

The study's insights into the role of application context in the effectiveness of homoglyph attacks reveal a critical consideration for developers and security professionals. The impact of fonts, input policies, and other contextual factors on the visibility and potential confusion caused by homoglyphs underscores the need for a holistic approach to security. It suggests that mitigating homoglyph attacks requires more than just technical solutions; it necessitates a comprehensive strategy that considers the user interface and experience aspects.

## 7 CONCLUSION

To conclude, our research has illuminated the critical issue of homoglyph attacks within the Ethereum Name Service (ENS) ecosystem. While ENS offers a powerful naming solution with its Unicode support, it also introduces security risks. Our contributions include identifying inconsistencies in application normalization processes and proposing the new attack scenarios that enlarges the attack surface in ENS. These findings underline the importance of robust normalization in ENS and similar systems. Collaborative efforts are essential to secure naming systems in the face of evolving threats.

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

# A PRELIMINARY STUDY

In summary, ENSIP-15 specifies how to tokenize Unicode characters, normalize tokens, and validate tokens. ENSIP-15 contains a lot of detailed rules, with two predefined lists (*spec.json* and *nf.json*) containing all valid characters and form information. For example, U+FE0F will be striped for emoji tokens during the normalization. When normalizing tokens, ENSIP-15 uses NFC (normalization form C as defined in Unicode standard [6]). When validating tokens, the community has a large, predefined map [3] to specify which character is not allowed, and which character should be replaced.

To understand how the normalization is implemented in ENS, we first analyze the popular libraries that support ENS. From ENS official website, we found a list of ENS libraries for various languages and we choose the first library for each language. Considering the popularity of JavaScript in Web3, we did manual search and found web3.js and ethers.js are the most popular libraries in Ethereum. In addition, in our preliminary study, we found many applications/libraries are based on eth-ens-namehash, which was recommended by the ENS official website as the normalization library. Hence, we added it into our list after our preliminary study. The collected libraries are written in different languages, including Java, Python, Go, and JavaScript, as listed in Table 3. It is worth noting that though 1.8.8 has 0 inconsistent domain normalization results with the official library, there do exist different normalization rules in them. One example we identified by manually checking the code is that U+24FF is no longer allowed since 1.9.0. The number of inconsistency results is 0 because there is no domain in our dataset using this character. Since 1.9.0, all dingbat digits are not allowed [7]. Web3.py has its own normalization implementation, which looks like the same code from ens-normalize 1.8.8 but written in Python. Hence, it has the same inconsistency as discussed above. ensjs is the official library developed by the ENS developers, which is also the most popular one, used by *ens.domain* (the official ENS domain registration website), Metamask (one of the most popular wallet), Alchemy (one of the most popular providers), etc. Since the goal of normalization is to reduce the usage of homoglyph characters in domains, we tried all the homoglyph domains detected

in Section 4 and checked whether all libraries can generate the same normalization results. After testing all homoglyph domains, we found that 4,330 domains are resolved with different results using different libraries. The different results could be two different addresses or one address with another one valid.

**Table 3: Evaluated Libraries.**

| Name | Weekly Downloads | Version | Language | Normalization Process Library | # of Inconsistent Domain Normalization Results |
|---|---|---|---|---|---|
| ensjs(Official) | 36,864 | 3.4.3 | JavaScript | ens-normalize 1.9.0 | Official |
| ethereum-ens | 13,778 | 0.8.0 | JavaScript | eth-ens-namehash | 4330 |
| eth-ens-namehash | 292,578 | 2.0.8 | JavaScript | idna-uts46-hx | 4330 |
| ~(Fork)* | 9,419 | 2.0.15 | JavaScript | eth-ens-namehash | 4330 |
| ethjs-ens | 1,663 | 2.0.1 | JavaScript | eth-ens-namehash | 4330 |
| ethers.js | 1,002,424 | 6.11.0 | JavaScript | ens-normalize 1.9.2 | 234 |
| web3.js | 474,896 | 4.4.0 | JavaScript | ens-normalize 1.8.8 | 0 |
| web3j | N/A | 4.10.3 | Java | adraffy/ENSNormalize.java 0.1.2 | 0 |
| KEthereum | N/A | 0.86.0 | Kotlin | Provider-based | N/A |
| web3.py | N/A | 6.15.0 | Python | web3.py | 0 |
| go-ens | N/A | 3.6.0 | Go | golang.org/x/net/idna | 4330 |
| ethereal | N/A | 2.9.0 | Go | go-ens | 4330 |
| delphereum | N/A | N/A | Pascal | Provider-based | N/A |

~(Fork)*:The project is forked from eth-ens-namehash. We use it here because it is recommended in the ENS documentations