# OpenReview forum: "Cutting through the Confusion: A Measurement Study of Homograph Domains in Ethereum Name Service"
_ACM.org/TheWebConf/2025/Conference — WWW 2025 Poster_

### Official Review · Reviewer_Z4i6 · 2024-11-15

**Novelty:** 4
**Technical Quality:** 3

**Review:**

I appreciate this research line. Although homoglyph attacks have been extensively discussed in the context of DNS, I'm not aware of work which investigated these issues for ENS. I think that this work deals with an important problem, but the technical execution and the quality of the presentation should be significantly improved before publication.

## Technical concerns

* Data collection: it is unclear to which extent the chosen dataset is comprehensive and representative. The paper mentions that the dataset includes 13 libraries and 264 apps for evaluation. How the libraries and the apps have been selected is not discussed, so it is impossible to assess possible sources of bias in the dataset construction. Did the authors scrape any public database of apps? How was their popularity measured? Most importantly, it doesn't seem that the dataset is publicly available, so it is even more difficult to understand whether its construction was performed with care.

* Unclear security implications: the paper discusses three inconsistencies and two possible attacks enabled by such inconsistencies. However, an inconsistency is just a bad programming practice / lack of compliance with published standards and does not necessarily enable an attack. The paper should more carefully discuss exploitation. Once an inconsistency is found, the analysis pipeline should use fuzzing or simply require human intervention to confirm that an attack is possible. The paper reports a single case study (MathWallet), for all the other apps it just reports numbers. 214 out of 264 apps (~80%) suffer from some inconsistencies, does it mean that all of them are vulnerable to some attack? I'm afraid not. I also do not understand to which extent authors analyzed libraries and what are the findings there: the paper just reports that 11 out of 13 libraries include some inconsistencies. Again, does it mean that any app using such libraries is vulnerable? Or are just some configurations of the libraries vulnerable? Are the inconsistencies exploitable and under which conditions? This would be very important to understand, given that multiple apps may use the same vulnerable library.

* Questionable methodology: I do not understand the choice of using dynamic analysis. Although I appreciate that the authors can hook functions and analyze invocations where domain names are passed as an argument, it seems to me that they are losing valuable information. Dynamic analysis is incomplete and requires a robust test generator to ensure appropriate coverage of the application code. How were tests generated? Conversely, invocations with domain names as an argument may have nothing to do with domain resolution, e.g., the domain resolution function may be part of a complicated stack trace starting from a network communication API: how can one discriminate the use cases of interest? Moreover, the paper never clarifies whether there is any reference implementation that may be leveraged to detect inconsistencies. The differential analyzer component of the pipeline compares the results of remote and local queries, but what is the ground truth?

* Lack of reproducibility: data and code do not seem to be available, which prevents an in-depth analysis of the artifacts. This also makes this research impossible to reproduce at this stage.

## Presentation issues

* Insufficient background information: the paper briefly explains some specific features of ENS, but does not provide enough context to understand the setting. For example, it does not mention the relevant standards and specifications, nor it clarifies why the hashing mechanism underlying domain registration is dangerous.

* Missing references: the paper mentions standards like ENSIP and UTS46 without appropriate bibliographic references.

* Missing examples: when presenting the normalization process, the paper would greatly benefit from an example to clarify the different steps. Also, the example should be reused later on to clarify the different types of inconsistencies that the authors are interested in. Without a concrete example, it is difficult to understand the bugs analyzed by this research.

**Questions:**

* Can you explain the dataset construction more in detail?
* Can you clarify which inconsistencies are exploitable and under which conditions?
* Can you motivate the choice of dynamic analysis and explain how do you generate tests?
* Can you make your code and data available for review?

**Reviewer Confidence:**

3: The reviewer is confident but not certain that the evaluation is correct

**Scope:**

3: The work is somewhat relevant to the Web and to the track, and is of narrow interest to a sub-community

---

### Official Review · Reviewer_22AZ · 2024-11-24

**Novelty:** 5
**Technical Quality:** 4

**Review:**

This paper serves as a good base for the discussion on the issue of ENS normalization.
I like the highlight of the severity of homoglyph attacks in ENS (direct) vs DNS (indirect).

I consider the method to automatically detect and analyze ENS normalization functions as sound, and it covers a lot of cases (e.g. server-based normalization).

While the method is thorough and extensive, the analysis based on it is quite light and does not yield many deep insights. More extensive analysis would have strengthened the work, e.g., on which libraries cause inconsistencies or whether all homoglyph characters are equally affected.

It was unclear to me how the selection of apps in Table 2 was made.

I was a bit surprised about the choice to only go for mobile apps (4.1). For a thorough analysis, I would have expected that the analysis also covered desktop apps. It would have made for an interesting comparison.

In a revised draft, I would recommend a re-read for grammar and spelling, especially in Section 4.

I appreciate that the authors went through responsible disclosure.

**Questions:**

* How was the selection of apps in Table 2 was made?

**Reviewer Confidence:**

3: The reviewer is confident but not certain that the evaluation is correct

**Scope:**

3: The work is somewhat relevant to the Web and to the track, and is of narrow interest to a sub-community

---

### Official Review · Reviewer_MmVy · 2024-11-25

**Novelty:** 4
**Technical Quality:** 4

**Review:**

The paper "Beyond Visual Confusion: Understanding How Inconsistencies in ENS Normalization Facilitate Homoglyph Attacks" addresses a pertinent issue within the Ethereum Name Service (ENS) ecosystem, focusing on the security vulnerabilities stemming from inconsistencies in the normalization process and the exploitation of Unicode homoglyphs.



**Quality:**
The paper presents a thorough analysis of the security vulnerabilities introduced by the use of Unicode in the Ethereum Name Service (ENS), particularly focusing on homoglyph attacks. The quality of the research is high, as evidenced by the systematic evaluation of the ENS normalization process and the identification of inconsistencies across various applications. The paper also contributes a novel tool for detecting application-level discrepancies in domain normalization.

**Clarity:**
The paper is well-structured. The introduction provides necessary background on ENS and motivates the study with new threats in ENS. The subsequent sections logically progress from background to methodology, results, and discussion, making the paper easy to follow. The use of figures and tables to illustrate homoglyph attacks and the workflow of ENS enhances the clarity of the presentation.

**Originality:**
The paper's originality lies in its focus on the ENS ecosystem and the specific challenges posed by Unicode normalization. While homoglyph attacks are not new, the paper provides a novel analysis of how these attacks manifest in the context of blockchain-based naming systems. The identification of new attack scenarios and the tool for detecting normalization inconsistencies contribute to the originality of the work.

**Significance:**
The significance of this work is underscored by the growing reliance on ENS in the Web3 ecosystem. With the increasing number of users and applications依托于ENS, the security of the system is paramount. The paper's findings have immediate implications for the integrity and security of ENS domains, potentially affecting millions of users. The paper also highlights the need for standardized normalization processes, which is a critical step towards mitigating homoglyph attacks.

**Pros:**
- Comprehensive analysis of ENS normalization and its vulnerabilities.
- Development of a tool for detecting normalization inconsistencies without source code access.
- Clear structure and logical progression of content.
- High relevance to the security of the ENS ecosystem and Web3 applications.
- Identification of new attack scenarios with significant implications for user security.

**Cons:**
- The paper could benefit from a deeper discussion on potential solutions or mitigation strategies beyond the identification of the problem.
- While the tool for detecting inconsistencies is mentioned, a more detailed description of its functionality and effectiveness would be beneficial.

**Questions:**

1. Could the authors provide more insights into the potential mitigation strategies that could be implemented at the protocol level?

**Reviewer Confidence:**

2: The reviewer is willing to defend the evaluation, but it is likely that the reviewer did not understand parts of the paper

**Scope:**

3: The work is somewhat relevant to the Web and to the track, and is of narrow interest to a sub-community

---

### Official Review · Reviewer_FeZZ · 2024-11-30

**Novelty:** 5
**Technical Quality:** 5

**Review:**

In this paper, the authors proposed a novel approach to detect application-level discrepancies in domain normalization without requiring access to the source code of the applications. To evaluate the performance of their approach, they analyzed over 200 real-world Web3 applications for inconsistencies in the ENS (Ethereum Name Service) normalization process. Additionally, the authors introduced new attack scenarios that expand the attack surface of ENS, emphasizing the urgent need to improve the security of naming systems.

However, my concerns regarding this survey are as follows:
1.	The authors might need to pay attention to the difference between the paper’s titles in the submission and the uploaded file.

2.	The novelty of the proposed approach seems insufficient. The authors introduced the normalization process in ENS and explored three sources of inconsistencies in the implementation of normalization in ENS. However, the detailed approach regarding how to identify the inconsistencies are not presented in an understandable manner, which leads to the novelty of this paper being insufficient.

3.	In the discussion of homoglyph attacks in ENS and DNS, the authors note that DNS attacks are mitigated to some degree through mechanisms like warnings or second-time confirmations. The authors also highlight the challenges of applying those methods in ENS, since victims often directly initiate transactions with provided addresses. However, the authors do not explore whether existing approaches for directly verifying domain name or address legitimacy in DNS could be adapted into ENS. If such adaptations are not feasible, or if no equivalent solutions exist, the authors might need to explicitly clarify this.

4.	While the authors introduce new attack scenarios for ENS, there is limited analysis of their practical impact. For example, the frequency of such attacks, their potential success rates, and their cost implications for attackers and defenders are not discussed. A more in-depth evaluation of these aspects would enhance the contributions.

**Questions:**

As in the review comments.

**Reviewer Confidence:**

2: The reviewer is willing to defend the evaluation, but it is likely that the reviewer did not understand parts of the paper

**Scope:**

3: The work is somewhat relevant to the Web and to the track, and is of narrow interest to a sub-community

---

### Official Review · Reviewer_FfKc · 2024-12-03

**Novelty:** 2
**Technical Quality:** 3

**Review:**

Homoglyph attacks replace or insert special characters in domain names. Those attacks can mislead the victims and resolve to malicious wallet addresses. To combat homoglyph attacks, Ethereum Name Service has employed a normalization process to check compliance with the ENSIP-15 standard before domain resolution. However, not every crypto application has correctly enforced this normalization process, causing inconsistencies in their and the official implementation. In this study, the authors observed that more than 60% of digital wallets and dApps show different extents of inconsistencies, leaving potential vulnerabilities to attackers.

Strengths
+ The study is well-motivated
+ The experiment setup is clearly-illustrated

Weaknesses
- Lack of in-depth practical implications

I would like to thank the authors for submitting their paper. I completely agree that homoglyph attacks are a simple yet dangerous threat in the context of ENS, particularly because they have a shorter attack pathway compared to DNS. The authors have put in considerable effort to collect data from major dApps on the market, and their findings highlight critical vulnerabilities in these applications.
My main concern lies in the practical implications of these findings. Given that a mature standard ENSIP-15 has already been developed, it seems that if dApps were to adopt a uniform and up-to-date standard for normalization, the inconsistency problem could be effectively solved. Consequently, this paper comes across more as a technical report highlighting some bugs rather than addressing a fundamentally challenging problem that could inspire future research efforts. Therefore, the primary audience for this work may be dApp developers rather than researchers focusing on ENS-related topics. A more thought-provoking and impactful topic could further uncover the deficiencies in the current ENSIP-15 standard and propose an improved version. Such a topic would carry greater significance and potentially appeal to a broader audience, including both researchers and developers working on ENS-related challenges.

**Questions:**

Is it possible that one App does not have any remote normalization query, but only local normalization queries? How to detect inconsistencies in this scenario?

**Reviewer Confidence:**

4: The reviewer is certain that the evaluation is correct and very familiar with the relevant literature

**Scope:**

4: The work is relevant to the Web and to the track, and is of broad interest to the community